# The Clinical Frailty Scale: Do Staff Agree?

**DOI:** 10.3390/geriatrics5020040

**Published:** 2020-06-25

**Authors:** Rebekah L. Young, David G. Smithard

**Affiliations:** 1Newham University Hospital, Bart’s Health NHS Trust, London E13 8SL, UK; 2Queen Elizabeth Hospital, Lewisham and Greenwich NHS Trust, London SE18 4QH, UK; david.smithard@nhs.net; 3Department of Sports Science, University of Greenwich, London SE10 9BD, UK

**Keywords:** frailty, clinical frailty score

## Abstract

The term frailty is being increasingly used by clinicians, however there is no strict consensus on the best screening method. The expectation in England is that all older patients should have the Clinical Frailty Scale (CFS) completed on admission. This will frequently rely on junior medical staff and nurses, raising the question as to whether there is consistency. We asked 124 members of a multidisciplinary team (consultants, junior doctors, nurses, and allied health professionals; physiotherapists, occupational therapists, dietitians, speech and language therapists) to complete the CFS for seven case scenarios. The majority of the participants, 91/124 (72%), were trainee medical staff, 16 were senior medical staff, 12 were allied health professions, and 6 were nurses. There was broad agreement both between the professions and within the professions, with median CFS scores varying by a maximum of only one point, except in case scenario G, where there was a two-point difference between the most junior trainees (FY1) and the nursing staff. No difference (using the Mann–Whitney U test) was found between the different staff groups, with the median scores and range of scores being similar. This study has confirmed there is agreement between different staff members when calculating the CFS with no specific preceding training.

## 1. Introduction

Frailty can be described as a clinical state in which the ability of older people to cope with every-day or acute stressors is compromised by increased vulnerability due to age-associated declines in physiological reserve and function across multiple organ systems [1]. It is estimated that around 10% of people aged over 65 years are frail, which increases to 25–50% of those aged over 85 [2]. Frailty can also be used to describe certain physical changes, such as muscle wasting and weakness, leading to reduced walking ability. The identification of these patients allows us to start a care pathway to address the issues contributing to frailty and avoid adverse outcomes.

All older people admitted to hospital should undergo a comprehensive geriatric assessment (CGA). It is recommended that this should commence on the day of admission [3]. An assessment of frailty is one component of the CGA, and similarly should be completed at the earliest opportunity. Assessments of frailty (frailty scales) are numerous [4] and rely on the recall of information, either by the patient or carer. The term frailty is being increasingly used by clinicians, however there is no strict consensus on the most appropriate screening scale [4]. 

The Clinical Frailty Scale, first described by Rockwood et al. in 2005 [5], is a nine-point scale where the assessor makes a judgement about the degree of a person’s frailty based upon clinical assessment and has been adopted by the Acute Frailty Network in the UK. The advantage the CFS has over other scales is that it offers a pictorial representation with a small description and is quick and simple to administer.

As the various frailty scores measure slightly different things, it is possible to score as severely frail on one and moderately frail on another. It is therefore important that the same scale is used throughout any one service.

The expectation in England is that all older patients should have the Clinical Frailty Scale completed at the time of admission or soon after [6]. This will frequently rely on the junior trainee medical staff and nurses. This raises the question as to whether there is consistency in completing the assessments.

## 2. Methodology

The participants (consultants, junior doctors, nurses, and allied health professionals; physiotherapists, occupational therapists, dietitians, speech and language therapists) were approached on the ward and at time of clinical education sessions/conferences. A total of 124 people agreed to take part (Table 1). They were provided with seven clinical case scenarios (Table 2) based on actual clinical scenarios and asked to provide a frailty score by referring to the Clinical Frailty Scale (1–9) (Figure 1). The results were completed anonymously; participants were requested to provide their profession and grade where appropriate. The participants had no or limited experience using the CFS, nor was any training provided on how to use it. The nurses and allied health professionals were of varying levels of qualification, from newly qualified to senior staff.

## 3. Results

The majority of participants—91/124 (72%)—were trainee medical staff, 16 were senior medical staff, 12 were allied health professions, and 6 were nurses (Figure 2).

There was broad agreement both between the professions and within the professions, with median CFS scores varying by a maximum of only one point, except in case scenario G, where there was a two-point difference between the most junior trainees (FY1) and the nursing staff (Figure 3). No difference (using multiple Mann–Whitney U) was found between different staff groups (basis between any two groups), with the median scores and range of scores all being very similar. 

## 4. Discussion

The severity of prior frailty at the time of admission is a prognostic indicator of outcome (length of stay, institutionalisation, and mortality) from acute medical and surgical illness [7,8,9,10]. Holistic medical management uses information from many sources, one of which is a frailty scale. For any tool to be adopted into clinical practice, it needs to be simple and quick to use. The CFS meets both criteria (Figure 1) and is used widely used in many geriatric services in England; however, the CFS, like any other assessment tool, needs to be consistent in identifying and grading frailty between clinical staff and between clinical services.

In a study conducted by a university-associated tertiary hospital in Melbourne, Australia, all patients aged 65 and over admitted to the general medical unit during August and September 2013 had their baseline CFS score documented by a member of the treating medical team [11]. Despite the lack of prior training for medical staff on the use of the CFS, increasing frailty was correlated with functional decline and mortality, supporting the validity of the CFS as a frailty screening tool for clinicians. This study, however, did not compare the scores between staff groups.

In a retrospective note review by a medical student, a CFS was completed and then compared to one completed by a nurse specialist during a comprehensive geriatric assessment. The agreement between the two assessments, using Cohen’s Kappa, was 0.63 [12]. An ICU-based study compared medical students with ICU doctors completing the CFS during patients’ stays and again found an agreement of 0.64 [13]. Rolfson et al. found a good interrater reliability with the Edmonton Frailty Scale completed by Geriatric specialist nurses [14]. 

In this study, the largest disagreement was with case scenario G, where there was a two-point difference between the most junior trainees (FY1) and the nursing staff. This could be explained by the fact the patient was independent with activities of daily living, but also suffering from falls, indicating that she may need more help. Overall, there was broad agreement, and therefore the CFS can be documented on patient admission and we can be reassured of the score’s consistency, despite it being used by different staff groups. The routine identification of frailty is good practice, as the identification of these patients allows us to start a care pathway to address the issues contributing to frailty and avoid adverse outcomes.

## 5. Conclusions

This study has confirmed that there is agreement between different staff members when conducting the CFS with no specific preceding training.

## Figures and Tables

**Figure 1 geriatrics-05-00040-f001:**
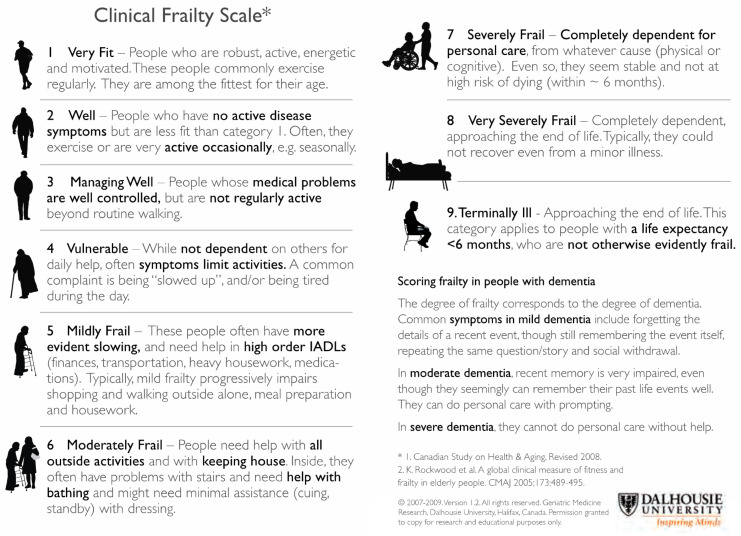
The Clinical Frailty Scale (CFS) [5].

**Figure 2 geriatrics-05-00040-f002:**
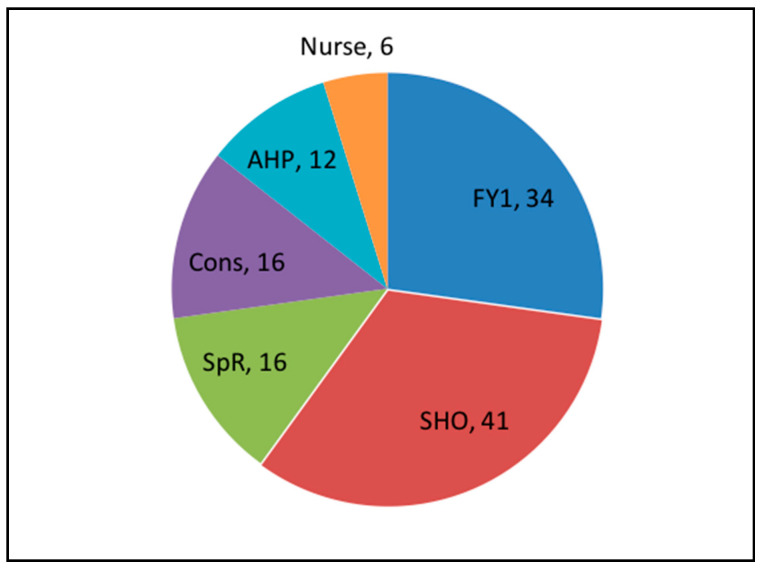
Distribution of staff completing the study.

**Figure 3 geriatrics-05-00040-f003:**
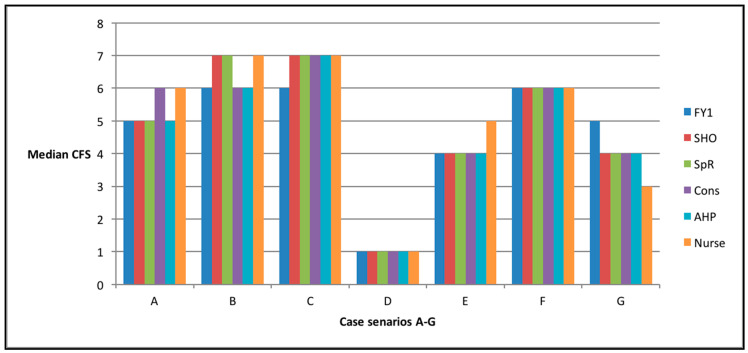
Chart to show the median CFS score calculated for each case scenario, divided by the member of the multidisciplinary team completing the score.

**Table 1 geriatrics-05-00040-t001:** Description of staff groups and previous experience with the CFS.

Staff Grade	Years Qualified	Experience with CFS
FY1	Immediately post qualification	None
SHO	Second year post training (may be longer depending on the individual)	None
Registrar	Min 4 years after undergraduate training	A few may have used the CFS
Consultant	At least nine years post graduate training	Depending on Specialty. Geriatricians would have experience other specialties not
Nursing/ AHP	Mixed 1–20	No exposure to CFS

**Table 2 geriatrics-05-00040-t002:** Clinical case scenarios.

	Case History
A.	84-year-old male. Admitted with a fall. Lives alone. Independent washing and dressing. Uses a walking stick in the house, housebound. Problems with urinary incontinence and wears pads. Has a BD care package when son away.
B.	81-year-old female. Walks with a Zimmer frame. Single level living. Undertakes a strip wash. Needs help with dressing, cooking, cleaning, shopping. Housebound. Carers 4 times a day. Unable to manage finances.
C.	91-year-old male. Independent with transfers from bed to chair but help otherwise to transfer chair to commode. Walks with a Zimmer frame but needs assistance. Help with personal activities of daily living (ADLs) (washing, dressing, shaving). Continence is an issue. Short term memory problems. Housebound. Unable to manage finances.
D.	74-year-old female. Working in an office, independent and self-caring. Drives a car. No care issues.
E.	89-year-old female. Walks with a stick and uses a 4-wheel shopper. Beginning to struggle with transfers (out of chair, off toilet) and lower half dressing. No package of care.
F.	84-year-old female. Recurrent falls and troubles with medication. Housebound, carers three times a day. Continent. Help with cooking, shopping and dressing. Requires help with medication. Cannot manage finances.
G.	82-year-old female. Falls, dementia. Independently mobile. Out shopping. Walks with a stick. Independent with personal and extended ADLS.

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
