# Peer review of "The Clinical Frailty Scale: Do Staff Agree?"

_geriatrics, 2020, doi:10.3390/geriatrics5020040_

Round 1
Reviewer 1 Report
Overall, this paper addresses an interesting topic, the agreement between different medical professionals in using the clinical frailty scale. I would like to rise some comments that will enhance the readability of the paper.
Introduction:
I suggest adding a definition of frailty in the first paragraph (and mentioning that there is not consensus on how to define frailty).
The clinical frailty scale should be introduced in the third paragraph before discussing its score. Also I recommend mentioning its advantage over other frailty tools.
“Mildly frail” was written twice in the third paragraph.
“The expectation in England is that all older patients should have the Clinical Frailty Scale“ is this a part of guidelines?? Also, I do not think that you need to use a reference here for CFS.
Methodology:
How seniority was established ? If by number of years, provide that in text.
Labels for groups should be provided (Figure 2 and 3)
I recommend matching the clinical scenarios numbering in Figure 1 and 3.
What is the basis of using Mann-Witney U test (given the presence of 6 groups)
Discussion:
I recommend providing some clinical implications of the results and explaining the two point difference between the nurses and the junior trainees.
Reviewer 2 Report
Thank you for the opportunity to review this study. The content would be of interest as a poster or conference paper but the paper would need to be further developed to merit publication. Please also consider the following points.
Title – The Clinical Frailty Scale: Can untrained staff use it?
The study participants may not have had specific training on the CFS but all were postgraduate clinicians – they were not untrained staff. Please rephrase
It describes those who are vulnerable and at a higher risk of adverse outcomes, for example those who are frail may not recover from minor illnesses.
By it presume you mean frailty – please state that. May not recover is a bit stark – perhaps replace with may not fully recover
The term frailty is being increasingly used by clinicians, however there is there is no strict consensus on its definition [4]
This is a very old reference. There is a commonly accepted definition used by WHO – suggest refer to this at least.
The score produced by the Clinical Frailty Scale (CFS) [5] can be used to divide people into several groups from pre-frail, mildly frail, mildly frail or severely frail.
The Clinical Frailty Scale is a continuum and does not use the term pre-frail. This is still a rather contentious term. Suggest stick with the terminology of the CFS categories
Methods: Which ward / specialty were the participants recruited from – their clinical experience is relevant in this study even if they had not been specifically trained to apply the CFS.
The participants are predominantly medical staff with less than 15% from a nursing / AHP background. If the authors intended to compare scoring by disciplines it would have been helpful to have more balanced groups. There is no information on the seniority of nurse / AHPs yet the medical staff are grouped by seniority.
Ref 3 has a typo - the cute medical unit.
Table 1 descriptors are variably described as clinical scenario / case study - be consistent with use of the word scenario
Figure 3 uses A-G but the scenario are labelled 1 – 7
In the discussion, the 2016 study in Australia is referenced but although this study sounds similar, the authors make no attempt to compare / contrast their findings.
What are the implications for practice or for training? What is the main message?
Round 2
Reviewer 1 Report
All my comments were addressed. The manuscript has greatly improved.
Author Response
Many thanks for your comments
Reviewer 2 Report
Thank you for revising your paper in line with the comments
However there is still no information on the clinical specialty experience of the participants or discussion of how representative they are of the front door clinical team likely to be completing the CFS in ED and admission wards.
Author Response
1. Sentence added on participants having no or limited experience of using the CFS.
2. Sentence added on participants having no training provided on how to us the CFS
3. Sentence added on nurses having varying levels of qualification from newly qualified to senior ward staff.
4. Box added to explain grades and CFS experience for clinical staff